Intraspecific variation of phragmocone chamber volumes throughout ontogeny in the modern nautilid Nautilus and the Jurassic ammonite Normannites

Tajika Amane 1 amane.tajika@pim.uzh.ch
Morimoto Naoki 2
Wani Ryoji 3
Naglik Carole 1
Klug Christian 1
1 Paläontologisches Institut und Museum, Universität Zürich , Zürich , Switzerland
2 Laboratory of Physical Anthropology, Graduate School of Science, Kyoto University , Kyoto , Japan
3 Faculty of Environment and Information Sciences, Yokohama National University , Yokohama , Japan
Wilson Laura
Electronic publication date: 2015 Oct 6
Publication date: 2015
Volume: 3
Electronic Location ID: e1306
Received 2015 Jun 19; Accepted 2015 Sep 18
Copyright: © 2015 Tajika et al.
Copyright year: 2015
Copyright holder: Tajika et al.
License: This is an open access article distributed under the terms of the Creative Commons Attribution License, which permits unrestricted use, distribution, reproduction and adaptation in any medium and for any purpose provided that it is properly attributed. For attribution, the original author(s), title, publication source (PeerJ) and either DOI or URL of the article must be cited.
License URL: https://creativecommons.org/licenses/by/4.0/

Keywords: Ammonoidea, Nautilida, Intraspecific variability, Sexual dimorphism, Growth, 3D reconstruction, Jurassic, CT scan, Cephalopoda

Funding: Swiss National Science Foundation SNF 200020_132870 200020_149120 200021_149119 This study is supported by the Swiss National Science Foundation SNF (project numbers 200020_132870, 200020_149120, and 200021_149119). The funders had no role in study design, data collection and analysis, decision to publish, or preparation of the manuscript.

==============================
Nautilus remains of great interest to palaeontologists after a long history of actualistic comparisons and speculations on aspects of the palaeoecology of fossil cephalopods, which are otherwise impossible to assess. Although a large amount of work has been dedicated to Nautilus ecology, conch geometry and volumes of shell parts and chambers have been studied less frequently. In addition, although the focus on volumetric analyses for ammonites has been increasing recently with the development of computed tomographic technology, the intraspecific variation of volumetric parameters has never been examined. To investigate the intraspecific variation of the phragmocone chamber volumes throughout ontogeny, 30 specimens of Recent Nautilus pompilius and two Middle Jurassic ammonites (Normannites mitis) were reconstructed using computed tomography and grinding tomography, respectively. Both of the ontogenetic growth trajectories from the two Normannites demonstrate logistic increase. However, a considerable difference in Normannites has been observed between their entire phragmocone volumes (cumulative chamber volumes), in spite of their similar morphology and size. Ontogenetic growth trajectories from Nautilus also show a high variation. Sexual dimorphism appears to contribute significantly to this variation. Finally, covariation between chamber widths and volumes was examined. The results illustrate the strategic difference in chamber construction between Nautilus and Normannites. The former genus persists to construct a certain conch shape, whereas the conch of the latter genus can change its shape flexibly under some constraints.

Introduction

Ammonoids and nautiloids are well-known, long-lived molluscan groups, both of which faced devastation at the end of the Cretaceous, but with different responses: extinction versus survival. What these two groups have in common is the external conch, which makes them superficially similar. Because of that, a number of palaeontologists investigated the ecology and anatomy of living Nautilus as an analogy for those of extinct ammonites over the last decades (e.g., Collins, Ward & Westermann, 1980; Saunders & Landman, 1987; Ward, 1987; Ward, 1988). However, it was Jacobs & Landman (1993) who argued that, despite its superficial morphologic similarity, Nautilus was an insufficient model to reconstruct ammonoid palaeoecology, given their phylogenetic positions, which are distant within the Cephalopoda. This argument is now widely accepted (e.g., Warnke & Keupp, 2005). Whereas palaeoecology and evolution of ammonoids need to be discussed based on their own fossil record, those of modern Nautilus can be satisfactorily analogized to fossil nautilids, which have borne persistent conch morphologies throughout their evolution (Ward, 1980).

Molluscan conchs are not only exoskeletal structures but also retain a complete record of their ontogeny because of their accretionary growth. One of the most important apomorphic structures of cephalopods, the chambered part of their conch (phragmocone), was used by most cephalopods and is still used by some cephalopods as a buoyancy device. The ammonite phragmocone has been of great interest for palaeontologists, in order to reveal otherwise-obscure aspects of ammonite palaeoecology (Geochemical analyses: Moriya et al., 2003; Lukeneder et al., 2010; Stevens, Mutterlose & Wiedenroth, 2015; 2 dimensional analyses of septal angles: Kraft, Korn & Klug, 2008; Arai & Wani, 2012). Buoyancy had not been examined by quantifying phragmocone volumes due to the lack of adequate methods until modern scanning technique enabled to reconstruct complete ammonite empirical volume models (Lemanis et al., 2015; Naglik, Rikhtegar & Klug, 2015; Tajika et al., 2015). Unfortunately, all of these contributions included only one specimen per species due to the great expenditure of time needed for segmenting the image stacks. Conclusions from such limited studies may be biased if the examined specimens represent more or less extreme variants of one species (intraspecific variation). The life mode of living Nautilus is known to be essentially demersal, retaining their buoyancy as either roughly neutral when active or slightly negative when at rest (Ward & Martin, 1978), even though they change their habitat frequently via vertical migration (Dunstan, Ward & Marshall, 2011). The majority of Nautilus ecology research has included studies on anatomy, behaviour, and habitat, whereas geometry and volume of their phragmocones, which are similar to that of fossil nautiloids, have been examined less frequently (e.g., Ward, 1979; Hoffmann & Zachow, 2011). Investigation of the relationship between Nautilus conchs and their ecology could become a reference to examine the relationship between fossil cephalopods and their palaeoecology.

Multiple methods have been applied to reconstruct conchs of cephalopods including both fossilized and extant animals (Kruta et al., 2011; Hoffmann et al., 2014; Lemanis et al., 2015; Naglik et al., 2015; Tajika et al., 2015; for general aspects of virtual palaeontology, see Garwood, Rahman & Sutton, 2010; Sutton, Rahman & Garwood, 2014). Non-destructive computed tomography (CT) superficially appears to be the best suitable method because rare fossils can be analysed without destroying them. Medical scanners are often used, but they often yield insufficient contrast between conch and internal sediment or cement because these materials may have similar densities (e.g., Garwood, Rahman & Sutton, 2010; Hoffmann & Zachow, 2011). Furthermore, the resolution obtained from medical scanners is not adequate, specifically in such cases where accurate measurements of minute structures such as ammonite protoconchs (as small as 0.5 mm in diameter; e.g., Lemanis et al., 2015) are required. Fossil cephalopods are thus difficult materials to examine by this non-destructive method, but conchs of living cephalopods with no sediment filling can easily be reconstructed with a good resolution. Computed microtomography (µCT) is an alternative because it has a stronger beam, resulting in high resolution and thus better reconstructions. µCT-imagery produced using high energy levels has greater penetrative power but suffers from the lack of contrast, however, making the subsequent segmentation process difficult.

By contrast, Lemanis et al. (2015) presented the first successful attempt to reconstruct an ammonite ammonitella in detail. They scanned a perfectly preserved hollow ammonite using phase contrast tomography. Propagation phase contrast X-ray synchrotron microtomography (PPC-SR-µCT) was employed by Kruta et al. (2011) who reconstructed ammonite radulae in detail. The limited availability of the facility, heavy data load, and potential contrast problems discourage application of this method for fossil cephalopods. In contrast to the non-destructive methods, destructive grinding tomography can be used to reconstruct fossilized cephalopods (Naglik et al., 2015; Tajika et al., 2015). This method, which preserves colour information of the shells (aiding in segmentation), does not require hollow preservation of fossils, thus permitting the examination of all well-preserved fossils without suffering from noise such as beam hardening or poor contrast, which commonly occur when using CT.

Volumetric analyses of intraspecific variability of phragmocone chambers throughout ontogeny have not previously been analysed in either Nautilus or ammonoids. Such data may contribute to the better understanding of the palaeoecology of extinct ammonoids and nautiloids. The aims of this study are to answer the following questions based on empirical 3D models reconstructed from real specimens: (1) How did chamber volumes change through ontogenetic development of ammonites and nautilids? (2) How much did the volumetric growth trajectories differ between two conspecific ammonites (exemplified using middle Jurassic Normannites)? (3) What was the intraspecific variation of volumetric growth trajectories of modern Nautilus? (4) Are the differences in chamber volumes between male and female nautilids significant? (5) Is there a difference in construction of chambers between the ammonites and modern Nautilus?

Material

Two ammonite specimens examined are from the Middle Jurassic and belong to the genus Normannites (Normannites mitis). One of them (Nm. 1) was reconstructed by Tajika et al. (2015) to test its buoyancy. Both specimens were found in the Middle Bajocian (Middle Jurassic) of Thürnen, Switzerland. The nicely preserved specimens are suitable for 3D reconstruction, even though one of the specimens (Nm. 2) has an incomplete aperture, which does not allow for buoyancy calculation. The maximum conch diameters of Nm. 1 and Nm. 2 are 50.0 mm and 49.0 mm, respectively.

An additional 30 conchs of Recent Nautilus pompilius (21 adults: 12 males, 9 females; 9 juveniles) were also studied. All of the conchs were collected in the Tagnan area in the Philippines (see Fig. 1 in Wani, 2004; Fig. 1 in Yomogida & Wani, 2013). Maturity of Nautiilus was defined as bearing black band, or septal crowding (for mature modification of Nautilus see Klug, 2004). Males and females were differentiated based on previous studies: mature males have larger shells and a broader, rounder aperture than females (Stenzel, 1964; Haven, 1977; Saunders & Spinosa, 1978; Arnold, 1984). In the juvenile stage, however, the sex is indeterminable since the morphological differences of shells are not profound. The details of the specimens are summarized in Table 1. The specimens are stored in Mikasa City Museum, Hokkaido, Japan.

Table 1 Details of the studied specimens, Normannites mitis from the Middle Jurassic, Switzerland, and modern Nautilus pompilius from the Philippines.

Specimen number	Species	Maturity	Sex	Maximum diameter (mm)	Number of chambers	
Nm.1	Normannites mitis	Mature	Male	50	60?	
Nm.2	Normannites mitis	Mature	Male	49	59?	
7	Nautilus pompilius	Mature	Female	189	35	
8	Nautilus pompilius	Mature	Female	152	30	
10	Nautilus pompilius	Mature	Female	175	32	
11	Nautilus pompilius	Mature	Female	165	30	
12	Nautilus pompilius	Mature	Female	168	33	
15	Nautilus pompilius	Mature	Female	189	33	
16	Nautilus pompilius	Mature	Male	183	33	
17	Nautilus pompilius	Mature	Male	183	33	
20	Nautilus pompilius	Immature	Indet.	105	26	
23	Nautilus pompilius	Immature	Indet.	112	26	
30	Nautilus pompilius	Immature	Indet.	147	30	
31	Nautilus pompilius	Immature	Indet.	136	29	
32	Nautilus pompilius	Immature	Indet.	136	32	
33	Nautilus pompilius	Immature	Indet.	135	27	
34	Nautilus pompilius	Immature	Indet.	144	32	
35	Nautilus pompilius	Immature	Indet.	124	28	
36	Nautilus pompilius	Immature	Indet.	157	37	
38	Nautilus pompilius	Mature	Male	150	31	
39	Nautilus pompilius	Mature	Male	147	32	
40	Nautilus pompilius	Mature	Male	151	30	
41	Nautilus pompilius	Mature	Male	184	34	
42	Nautilus pompilius	Mature	Female	169	33	
43	Nautilus pompilius	Mature	Male	155	31	
44	Nautilus pompilius	Mature	Male	164	35	
46	Nautilus pompilius	Mature	Male	160	31	
48	Nautilus pompilius	Mature	Male	165	35	
51	Nautilus pompilius	Mature	Female	179	33	
53	Nautilus pompilius	Mature	Male	181	36	
54	Nautilus pompilius	Mature	Male	164	29	
56	Nautilus pompilius	Mature	Female	176	32	

Methods

3D reconstructions of ammonites

Grinding tomography was employed to reconstruct the two Jurassic ammonite specimens. This method has been applied to previous studies for invertebrates, e.g., bivalves (Götz, 2003; Götz, 2007; Götz & Stinnesbeck, 2003; Hennhöfer, Götz & Mitchell, 2012; Pascual-Cebrian, Hennhöfer & Götz, 2013) and ammonoids (Naglik et al., 2015; Tajika et al., 2015). During each of the 422 grinding phases, 0.06 mm was automatically ground off of the specimens until the specimen was completely destroyed. Subsequently, each ground surface was automatically scanned with a resolution of 2,400 dpi. Due to the very high number of slices and the very time consuming segmenting process, only every fourth scan of the obtained image stack was segmented. The voxel sizes of x, y and z dimensions are 0.025, 0.025 and 0.24 mm, respectively. We separately segmented the external conch, all septa, and the siphuncle manually using Adobe® Illustrator (Adobe Systems). The segmented image stacks have been exported to VGstudiomax® 2.1 (Volume Graphics), which produced 3D models out of the 2D image stacks. Further technical details for the ammonite reconstructions are given in Tajika et al. (2015) and for the general procedure of grinding tomography in Pascual-Cebrian, Hennhöfer & Götz (2013).

3D reconstructions of modern Nautilus

Conchs of all specimens were scanned at the Laboratory of Physical Anthropology of Kyoto University using a 16-detector-array CT device (Toshiba Alexion TSX-032A) with the following data acquisition and image reconstruction parameters: beam collimation: 1.0 mm; pitch: 0.688; image reconstruction kernel: sharp (FC30); slice increment: 0.2 mm; tube voltage and current: 120 kV 100 mA. This resulted in volume data sets with isotropic spatial resolution in the range of 0.311 and 0.440 mm. The obtained data sets were exported to Avizo® 8.1 (FEI Visualization Sciences Group) where segmentation was conducted. As mentioned in Hoffmann et al. (2014), the calculated mass of a specimen based on the CT data set does not correspond exactly to the actual mass measured on the physical specimen due to noise and the partial volume effect (PVE) from the scan, which may cause significant errors during the segmentation process. Wormanns et al. (2004) reported that segmentation can also introduce errors between specimens. In our scans, the resulting differences between the actual masses of the conchs and the calculated mass ranged from 50 to 63%. However, use of the same devices and methods and a combination of the same grey-scale threshold value for the outer whorls and the manual tracing for the innermost whorls reduce the noise and preserve the overall trend of variability in volumes between each specimen. Out of 45 scanned specimens, only 30 scanned specimens with nearly the same contrast were carefully chosen and analysed, while scans from other 15 specimens with different contrasts were discarded to minimize errors which may occur from differences in contrast between scans. Nevertheless, the variability is to some degree affected by the errors due to the noise and PVE. A list of the differences between calculated shell volumes and estimated actual shell volumes calculated from mass measurements is provided in Table S1 (estimated volume error: 60.8–91.3%). The segmented data sets were exported as STL files using the software Avizo® 8.1. The volumetric data from the phragmocone were extracted and calculated in Meshlab (ISTL–CNR research center) and Matlab 8.5 (Math Works), respectively. The measurements of the diameters and widths of the conchs were conducted with the program ForMATit developed by NM.

Results

Difference between two Normannites specimens in ontogenetic volume changes

Constructed 3D models of the ammonites are shown in Fig. 1 (1A–1D). Measured chamber volumes (Table 2) were plotted against chamber numbers (Fig. 2). In the two Normannites specimens, the overall trends of growth trajectories of individual chamber volumes (Fig. 2A) are more or less the same, showing logistic increase throughout ontogeny until the onset of the so-called ‘terminal countdown’ (Seilacher & Gunji, 1993) when they start showing a downward trend over the last 5 chambers (Nm. 1) and over the last 7 chambers (Nm. 2). The curve from Nm. 1 illustrates a nearly steady growth rate even though a syn vivo epizoan worm with mineralized tube grew on the fifth whorl of the ammonite (Tajika et al., 2015). By contrast, Nm. 2 does not show traces of any syn vivo epizoans, but it displays a sudden decrease of the volume of the 45th chamber where another trend sets off, which persists to the last chamber. In addition, we plotted the cumulative volumes of the phragmocone chambers against chamber numbers (Fig. 2B). Since the curves are derivatives of those of Fig. 2, the phragmocone volumes increase with the same trend. The cumulative phragmocone volume of Nm. 1 is larger than that of Nm. 2, although the latter retained the larger phragmocone volume throughout ontogeny until the onset of the morphologic countdown.

Figure 1 3D reconstructions of the two specimens of Normannites mitis, modern Nautilus pompilius (specimen 17), and their phragmocones.

(1A) 3D model of Normannites mitis (Nm. 1); (1B) 3D model of Normannites mitis (Nm. 2); (1C) extracted phragmocone of Nm. 1 (1D); extracted phragmocone of Nm. 2; (2A, B) 3D models of Nautilus pompilius (specimen 17); (2C) extracted phragmocone of Nautilus pompilius (specimen 17); (2D) Backface of 3D model of Nautilus pompilius (specimen 17). Scale bars are 1 cm.

Figure 2 Volumes plotted against chamber numbers in Normannites mitis. The volumes prior to chamber 25 (Nm. 1) and 27 (Nm. 2) have not been measured.

(A) Scatter plot of chamber numbers and individual chamber volumes; (B) Scatter plot of chamber numbers and cumulative phragmocone volumes.

Table 2 Raw data of measured chamber volumes and widths in Normannites mitis.

Normannites mitis	
Specimen	Nm. 1	Nm. 2	
Chamber	Volume (mm3)	Width (mm)	Volume (mm3)	Width (mm)	
25	0.9	–	–	–	
26	1.3	–	–	–	
27	2.0	–	1.6	–	
28	2.1	2.6	2.5	–	
29	2.6	2.6	3.0	–	
30	2.9	2.7	3.8	–	
31	3.4	2.6	4.8	–	
32	4.2	3.1	5.3	–	
33	6.0	4.1	7.4	–	
34	9.6	4.1	8.8	–	
35	8.6	4.6	11.3	–	
36	10.7	4.6	12.4	–	
37	12.9	4.6	16.2	3.9	
38	16.0	4.6	16.8	3.9	
39	16.2	4.7	20.4	4.8	
40	26.1	5.5	30.8	5.8	
41	28.9	5.8	43.1	7.2	
42	39.2	6.5	61.0	7.7	
43	49.7	7.4	72.4	7.7	
44	59.1	7.9	78.6	7.7	
45	66.7	8.4	54.0	7.2	
46	81.4	8.9	76.3	7.2	
47	99.4	9.4	93.1	7.9	
48	113.3	9.8	130.4	8.6	
49	155.1	10.3	198.6	11.0	
50	171.8	11.3	296.0	13.2	
51	255.9	12.5	380.5	15.1	
52	338.7	14.6	446.4	15.1	
53	397.6	15.1	458.6	15.1	
54	498.5	16.6	425.7	13.9	
55	557.4	16.6	384.6	13.4	
56	510.2	17.5	409.1	15.1	
57	576.1	17.5	428.5	15.4	
58	528.4	18.0	375.1	15.9	
59	497.3	18.0	339.3	15.4	
60	410.5	18.0	–	–	

Intraspecific variability of modern Nautilus in ontogenetic volume changes

Constructed 3D models of modern Nautilus are shown in Fig. 1(2A–2D). As in the Jurassic ammonite, individual chamber volumes and phragmocone volumes (Table 3) were plotted against chamber numbers (Figs. 3A and 3B). Figure 3 shows that all the curves increase logistically, as in the ammonites, with a rather high variability. As far as the terminal countdown is concerned, only the last or no chamber of the adult specimens shows the volume decrease. By contrast, the two ammonites show this decrease over the last 5–7 chambers (even higher numbers of chambers may be included in other ammonite species: e.g., 18 in the Late Devonian Pernoceras, 14 in the Early Carboniferous Ouaoufilalites; see Korn, Bockwinkel & Ebbighausen, 2010; Klug et al., 2015) bearing the irregular growth. It has been reported that mature males of Nautilus from the Fiji Islands have larger shells and a broader, rounder aperture than those of females (Stenzel, 1964; Haven, 1977; Saunders & Spinosa, 1978; Arnold, 1984) but there were no significant differences between sexes in shell form in Nautilus from the Philippines (Tanabe et al., 1983). In order to assess the differences between male and female conchs, their growth trajectories are shown in Fig. 4. Maximum diameters of the conchs versus number of chambers (Fig. 5A) and maximum diameters versus phragmocone volumes are also plotted (Fig. 5B) to assess if previously-recognized morphologic differences between males and females of Nautilus are detectable here. A statistical test (analysis of the residual sum of squares; ARSS) was carried out to determine whether there are differences between males and females in growth trajectories (Fig. 4B) and morphological features (maximum diameters of conchs vs. number of chambers; maximum diameters of conchs vs. phragmocone volumes; Figs. 5A and 5B). This test is used to compare linear models (Zar, 1996). A similar statistical test, which compares non-linear models, described by Akamine (2004) was also conducted for growth trajectories of males and females (Fig. 4C) to verify whether or not the results from ARSS are valid. The results of the statistical tests suggest that there are significant differences between males and females (Tables 5 and 6).

Figure 3 Chamber volumes plotted against chamber numbers in all examined Nautilus pompilius.

(A) Scatter plot of chamber numbers and individual chamber volumes; (B) Scatter plot of chamber numbers and phragmocone volumes.

Figure 4 Comparison between males and females. Chamber volumes plotted against chamber numbers in Nautilus pompilius.

Squares and diamonds represent the female and male, respectively. (A) Scatter plot of chamber numbers and individual volumes; (B) Semilog scatter plot of chamber numbers and individual volumes; (C) Scatter plot of chamber numbers and cumulative phragmocone volumes.

Figure 5 Comparison between males and females.

Squares, diamonds, and triangles represent the female, male, and indeterminable sex, respectively. (A) Scatter plot of maximum conch diameters and chamber numbers of a specimen; (B) Scatter plot of maximum conch diameters and the phragmocone volume.

Comparison of chamber formation between ammonites and Nautilus

Widths (for Normannites: Table 2; for Nautilus: Table 4) and volumes of each chamber were plotted against chamber numbers for the ammonites (Fig. 6) and Nautilus (Fig. 7). It should be noted that the widths of each chamber for the ammonites may not be very accurate. For instance, for the widths of the 42nd to 44th chamber of Nm. 2 (Fig. 6B), we obtained the same value (7.7 mm), which presumably does not represent the actual width. This has been caused by the reduction in resolution resulting from segmenting only every 4th slice with an increment between two images 0.24 mm in voxel z (instead of 0.06 mm; see the method section above for details). In addition to the low resolution, the obscure limit between chambers and septa at the edges of the chambers (on the flanks) in the slices might also have resulted in some errors in segmentation. However, the overall trend of the widths through ontogeny should still be correctly depicted and thus, the errors mentioned above were negligible for our study (Fig. 6B).

Figure 6 Volumes and widths of chambers plotted against chamber numbers in Normannites mitis. Squares and diamonds represent volumes and widths, respectively.

(A) Nm.1; (B) Nm. 2.

Figure 7 Volumes and widths of chambers plotted against chamber numbers in Nautilus pompilius.

Squares and diamonds represent volumes and widths, respectively. (A) Specimen 8; (B) Specimen 7; (C) Specimen 53. Specimens with different growth trajectories were analysed.

Table 3 Raw data of measured chamber volumes in Nautilus pompilius.

Nautilus pompilius	
Volumes (ml)	
Chamber	7	8	10	11	12	15	16	17	20	23	
1	0.0011	0.0080	0.0082	0.0118	0.0139	0.0088	0.0099	0.0101	0.0153	0.0120	
2	0.0123	0.0331	0.0257	0.0416	0.0384	0.0317	0.0145	0.0307	0.0329	0.0370	
3	0.0468	0.1013	0.0760	0.1056	0.1091	0.0866	0.0424	0.0882	0.0922	0.1440	
4	0.1142	0.1951	0.1539	0.1980	0.1809	0.1571	0.1109	0.1584	–	0.1904	
5	0.1837	0.2417	0.2028	0.2214	0.2050	0.2032	0.1859	1.9870	0.2939	0.1658	
6	0.2236	0.1264	0.1397	0.1244	0.1081	0.1327	0.2182	1.2660	0.1387	–	
7	0.1287	0.1987	0.1736	0.2603	0.1742	0.1711	0.1610	0.1911	0.1504	0.1875	
8	0.1767	0.2520	0.2027	0.2639	0.2046	0.1654	0.2183	0.2065	0.1695	0.2451	
9	0.2265	0.2800	0.2472	0.3593	0.2370	0.2352	0.2730	0.2418	0.2092	0.3563	
10	0.2619	0.3126	0.2873	0.4043	0.3378	0.2344	0.3047	0.2709	0.2314	0.3615	
11	0.3097	0.4201	0.3461	0.4913	0.3364	0.2671	0.3856	0.3332	0.3010	0.2962	
12	0.3254	0.5510	0.4246	0.5882	0.3992	0.3542	0.4402	0.4326	0.4017	0.5029	
13	0.3419	0.6398	0.4958	0.6988	0.4677	0.4407	0.5293	0.4632	0.3846	0.6454	
14	0.4342	0.8348	0.6386	0.9175	0.5496	0.5297	0.6218	0.5654	0.5069	0.7712	
15	0.5986	0.9723	0.7534	1.1123	0.7096	0.5844	0.7034	0.7108	0.5902	0.8968	
16	0.6954	1.1514	0.9129	1.2902	0.8697	0.6870	0.8370	0.8858	0.7431	1.0808	
17	0.7329	1.5420	0.9722	1.5716	0.9987	0.8377	1.1188	1.0799	0.9711	1.3026	
18	0.8595	1.8436	1.2630	2.0393	1.1376	1.0711	1.3181	1.3902	1.1740	1.5484	
19	1.1690	2.4328	1.6209	2.3768	1.4889	1.4076	1.6280	1.7581	1.5174	1.7800	
20	1.3495	2.8077	1.6611	3.1048	1.8336	1.6886	1.8692	2.2017	1.8071	2.4023	
21	1.7666	3.4284	2.2127	3.8014	2.2195	2.2858	2.3806	2.7137	2.2284	2.8600	
22	2.0429	4.7002	2.4138	5.1772	2.8784	2.6827	3.0621	2.9842	2.8115	3.4343	
23	2.6836	5.8684	3.6654	6.4984	3.4312	3.0022	3.8081	4.2956	3.3740	4.4262	
24	3.1432	7.3975	3.9932	6.3292	4.0784	3.9945	4.8836	5.7708	4.3020	5.5624	
25	3.8981	9.2433	5.9550	10.8780	4.8802	5.2016	6.4403	6.5720	5.5132	6.8422	
26	4.7613	12.1851	7.2257	13.0345	6.1415	6.9912	7.7378	8.3211	6.5154	8.3682	
27	6.2645	14.8837	9.1428	15.1136	7.1537	6.9741	10.2469	9.7510	–	–	
28	7.6362	18.9061	11.6261	15.0097	9.3969	9.9014	11.9939	12.6750	–	–	
29	8.9947	23.4334	14.3625	18.0443	11.4332	13.0762	15.4993	15.4005	–	–	
30	11.6532	21.7685	18.6543	16.2038	13.7770	15.9414	18.4287	17.8146	–	–	
31	14.3670	–	22.4427	–	17.3911	21.2605	21.4919	22.5759	–	–	
32	18.7249	–	25.6854	–	19.8835	25.8978	26.6814	25.5356	–	–	
33	22.7825	–	–	–	19.3914	23.7399	21.6118	29.6341	–	–	
34	28.9011	–	–	–	–	–	–	–	–	–	
35	25.0228	–	–	–	–	–	–	–	–	–	
36	–	–	–	–	–	–	–	–	–	–	
Chamber	30	31	32	33	34	35	36	38	39	40	
1	0.0009	0.0081	0.0015	0.0081	0.0076	0.0010	0.0216	0.0098	0.0106	0.0101	
2	0.0093	0.0307	0.0112	0.0138	0.0238	0.0151	0.0566	0.0283	0.0415	0.0413	
3	0.0491	0.1274	0.0372	0.0523	0.0673	0.0441	0.1162	0.0987	0.0610	0.1276	
4	0.1152	0.0900	0.1024	–	–	0.1044	0.1356	0.1778	0.1955	0.2445	
5	0.2002	0.1677	0.1703	0.2591	0.1836	0.1951	0.0903	0.2302	0.2274	0.2826	
6	0.2263	0.2333	0.2108	0.3325	0.0731	0.1551	0.0677	0.1288	0.1437	0.1377	
7	0.1298	0.1515	0.1059	0.1488	0.1445	0.1211	0.0875	0.1754	0.2137	0.1577	
8	0.2507	0.1968	0.1578	0.2810	0.1506	0.2130	0.1325	0.2319	0.2327	0.2791	
9	0.2457	0.2774	0.1513	0.3327	0.1912	0.2311	0.1384	0.2424	0.2748	0.3210	
10	0.3184	0.3346	0.2389	0.3967	0.2178	0.3198	0.1650	0.3559	0.3628	0.3354	
11	0.3811	0.4392	0.2743	0.4897	0.2891	0.3354	0.1998	0.3528	0.3506	0.4696	
12	0.4743	0.4943	0.2953	0.5830	0.2969	0.4166	0.2167	0.4391	0.4582	0.5265	
13	0.5728	0.5368	0.3519	0.6721	0.3613	0.4578	0.2776	0.5343	0.5336	0.6694	
14	0.6597	0.5660	0.4364	0.7652	0.4548	0.4956	0.3469	0.6659	0.5510	0.7933	
15	0.8527	0.6376	0.4978	0.9763	0.5328	0.6623	0.3984	0.8642	0.7349	0.9906	
16	0.9906	0.9415	0.5625	1.1348	0.6799	0.8069	0.4671	1.0654	0.8903	1.1742	
17	1.2034	1.2099	0.6816	1.5905	0.8066	0.9817	0.5594	1.2510	1.1273	1.4877	
18	1.5362	1.4315	0.8131	1.7629	0.9474	1.2012	0.7268	1.5251	1.3187	1.8743	
19	1.7694	1.7856	0.9522	2.2513	1.2071	1.3979	0.8601	1.8645	1.6630	2.3415	
20	2.0389	1.9788	1.1264	3.0569	1.4379	1.8163	0.9568	2.3037	2.1185	2.8293	
21	2.8880	2.6252	1.4726	3.5649	1.7398	2.2560	1.1435	3.0019	2.5387	3.4876	
22	3.3829	3.0792	1.5172	4.5086	2.0732	2.7278	1.3670	3.8435	3.1226	4.1792	
23	3.6387	4.1283	2.0698	5.8497	2.6354	3.5553	1.4716	5.0250	4.3051	5.2172	
24	5.5978	4.8777	2.5775	7.8330	3.0635	4.2451	1.9052	5.9666	5.0770	6.9681	
25	6.6551	6.6584	2.9776	10.0561	3.7968	5.6042	2.1254	7.4867	6.4071	9.1711	
26	8.4330	8.2790	3.7357	12.3302	4.6313	7.0547	2.4165	9.5045	7.9895	11.4558	
27	10.9828	10.7209	4.2277	16.8159	5.7833	8.7436	3.1417	12.3553	9.9455	14.8504	
28	14.0144	13.7381	5.9748	–	6.7042	11.2815	3.9028	15.4332	12.1152	18.7030	
29	17.9875	16.9861	6.9056	–	8.9703	–	4.0146	19.5149	16.8772	21.2875	
30	22.9906	–	8.7325	–	10.3012	–	5.5218	22.3363	19.1758	20.7897	
31	–	–	11.0929	–	13.7366	–	6.4224	21.7169	22.8448	–	
32	–	–	13.4910	–	16.1578	–	8.3757	–	10.9346	–	
33	–	–	–	–	–	–	9.7338	–	–	–	
34	–	–	–	–	–	–	13.6863	–	–	–	
35	–	–	–	–	–	–	15.1073	–	–	–	
36	–	–	–	–	–	–	19.3678	–	–	–	
Chamber	41	42	43	44	46	48	51	53	54	56	
1	0.0100	0.0054	0.0090	0.0050	0.0265	0.0047	0.0175	0.0061	0.0100	0.0093	
2	0.0292	0.0247	0.0306	0.0186	0.0771	0.0183	0.0470	0.0181	0.0342	0.0315	
3	0.0905	0.0708	0.0881	0.0496	0.1503	0.0468	0.1091	0.0549	0.0913	0.0873	
4	0.1417	0.1532	0.1587	0.1075	0.1971	0.0971	0.1735	0.1069	0.1690	0.1472	
5	0.2076	0.2127	0.2030	0.1600	0.1691	0.1455	0.1890	0.1296	0.1763	0.2053	
6	0.1124	0.1729	0.1402	0.1743	0.1699	0.1296	0.1049	0.0991	0.0946	0.2054	
7	0.1508	0.1493	0.1831	0.1235	0.2227	0.0904	0.1476	0.0782	0.2062	0.1376	
8	0.1697	0.2169	0.2357	0.1846	0.2459	0.1272	0.1975	0.1243	0.1836	0.1697	
9	0.2163	0.2819	0.2991	0.1938	0.3018	0.1317	0.2505	0.1579	0.2436	0.2927	
10	0.2786	0.3644	0.3365	0.2052	0.3498	0.1749	0.2403	0.1804	0.3114	0.3502	
11	0.3207	0.4320	0.3932	0.2967	0.4234	0.1962	0.3590	0.2276	0.3474	0.3969	
12	0.4028	0.5334	0.4842	0.3297	0.4885	0.2544	0.3641	0.2631	0.3622	0.4777	
13	0.3789	0.6502	0.5946	0.4074	0.6444	0.2892	0.4552	0.2786	0.4824	0.5308	
14	0.3697	0.8009	0.7316	0.4628	0.7167	0.3641	0.5052	0.3390	0.5973	0.7307	
15	0.4970	1.1199	0.8541	0.5346	0.9162	0.4755	0.6910	0.4319	0.7167	0.9280	
16	0.7079	1.3768	1.0209	0.6888	1.1237	0.5788	0.8284	0.5339	0.9275	1.0657	
17	0.8187	1.6980	1.3506	0.8180	1.4206	0.7132	0.9799	0.6473	1.0603	1.3458	
18	0.9482	2.1715	1.5373	0.9756	1.5012	0.7694	1.2509	0.7253	1.3217	1.4686	
19	1.1905	2.5023	1.9608	1.2337	2.1029	0.9727	1.4561	1.0164	1.5396	1.8512	
20	1.4391	3.1098	2.1780	1.5515	2.4645	1.2410	1.7334	1.0873	1.9675	2.3222	
21	1.7595	4.1807	2.9540	1.9814	3.2696	1.4992	2.1757	1.4246	2.4795	2.8080	
22	2.1740	5.2048	3.5435	2.6261	3.7837	1.9494	2.6698	1.6820	3.0712	3.4655	
23	2.6913	6.7107	4.6642	2.7189	4.6898	2.2113	3.5267	1.9744	3.6531	4.4481	
24	3.3197	8.3822	5.6355	4.1850	6.2850	2.6959	3.8889	2.5256	4.6271	5.2782	
25	3.9711	9.8258	7.2365	4.8333	7.7151	3.3410	5.4467	3.2210	5.7637	6.6173	
26	5.1796	14.0874	8.8481	6.3843	9.6012	4.1416	7.0138	3.7303	7.4533	8.4093	
27	6.3708	16.9760	10.8568	7.8972	12.4969	5.2332	8.5615	4.3930	9.1647	10.4171	
28	7.3239	20.3430	13.3318	10.4022	16.2270	6.3615	10.4667	4.8603	10.4041	13.1087	
29	9.5327	25.8620	16.3558	13.1177	19.5241	7.5145	13.5815	6.7250	13.7364	15.5874	
30	11.9083	24.6416	18.0790	17.3703	24.7367	9.4214	17.3426	8.8509	18.1738	20.3345	
31	14.4140	–	20.2377	20.7735	20.2453	12.4135	20.6539	11.0477	22.7498	22.5689	
32	18.5821	–	–	27.8035	–	15.0377	25.8738	14.1953	24.6066	19.6485	
33	23.3349	–	–	27.8442	–	18.3685	21.4921	17.2212	15.7064	–	
34	27.2882	–	–	–	–	22.6245	–	22.1384	–	–	
35	–	–	–	–	–	26.4088	–	26.0839	–	–	
36	–	–	–	–	–	–	–	21.8776	–	–	

Table 4 Raw data of measured chamber widths of Nautilus pompilius.

Nautilus pompilius	
	Widths (mm)	
Chambers	Specimen 8	Specimen 7	Specimen 53	
6	–	–	–	
7	–	–	–	
8	–	–	–	
9	–	–	–	
10	–	–	–	
11	13.8	–	13.8	
12	14.1	11.5	14.1	
13	14.5	12.4	14.5	
14	15.2	13.2	15.2	
15	16.3	14.2	16.3	
16	16.6	15.1	16.6	
17	17.4	16.3	17.4	
18	18.2	17.0	18.2	
19	19.3	17.8	19.3	
20	20.4	19.1	20.4	
21	21.8	20.4	21.8	
22	22.6	21.4	22.6	
23	24.6	22.9	24.6	
24	26.2	24.3	26.2	
25	30.0	26.1	30.0	
26	30.1	27.4	30.1	
27	32.3	29.2	32.3	
28	34.0	31.0	34.0	
29	36.2	33.1	36.2	
30	39.7	36.1	39.7	
31	42.4	38.9	42.4	
32	45.2	41.7	45.2	
33	48.3	44.7	48.3	
34	52.8	47.9	52.8	
35	55.6	51.5	55.6	
36	–	54.5	–	

Table 5 Results of statistical tests (analyses of the residual sum of squares) comparing linear regressions of males and female.

Comparison	N (male)	N (female)	RSS (male)	RSS (female)	DF (male)	DF (female)	t	Significance	
Chamber number vs. chamber volume (between the 1st and 5th chambers)	60	45	59.9	4,601	58	43	0.005	ns (P > 0.5)	
Chamber number vs. chamber volume (from the 6th chamber)	332	243	108.3	104.0	330	240	16.8	s (P < 0.05)	
Maximum diameter vs. number of chambers	12	9	46.5	14.6	10	7	1.9	s (P < 0.1)	
Maximum diameter vs. total volume of phragmocone	12	9	927.6	721.0	10	7	2.2	s (P < 0.1)	
Notes.

N number of samples

RSS residual sum of squares

DF degree of freedom

ns not significant

s significant

Table 6 Results of a statistical test (an analysis of the residual sum of squares) comparing nonlinear regressions of males and females.

Comparison	RSS (total)	RSS (male)	RSS (female)	DF (male)	DF (female)	F	Significance	
Chamber number vs. chamber volume (from the 6th chamber)	2775.3	1670.0	1040.4	332	243	4.55	s (P < 0.1)	
Notes.

RSS residual sum of squares

DF degree of freedom

ns not significant

s significant

Discussion

Ontogenetic volumetric growth of ammonites

Due to preservation and limited resolution, the chambers in the first two whorls of the Jurassic ammonites could not be precisely measured. Therefore, the chamber numbers referred to below were estimated based on recognizable chambers and more or less constant septal spacing. There appears to be a subtle point where the slope of the curves changes at around the 28–29th chamber (Fig. 2B), corresponding to a conch diameter of about 4.5 mm. This change may represent the end of the second growth stage of ammonoids, the neanic stage, because it has been reported that the neanic stage of ammonoids lasts until a conch diameter of 3–5 mm (Bucher et al., 1996). This point may have been related to the change of their mode of life, i.e., from planktonic to nektoplanktonic or nektonic (Arai & Wani, 2012). Taking this into account, the first two whorls of the conch comprise the first two growth stages, namely the embryonic and the neanic stages (Bucher et al., 1996; Westermann, 1996; Klug, 2001). Note that since the volumes of chambers formed before the 25th and 27th in Nm. 1 and Nm. 2 have not been measured due to the poor resolution, the transition between the first two growth stages has not been examined. Naglik et al. (2015) three-dimensionally examined three different Palaeozoic ammonoid species. They documented a change in the slope of growth trajectories around the 19th–21st chamber in each specimen. The last several chambers display fluctuating growth known as terminal countdown (Seilacher & Gunji, 1993). In Nm. 2, an abrupt decrease of chamber volume occurred at the 45th chamber, marking another trend resulting in a lower cumulative volume than in Nm. 1. It is known that injuries can affect the septal spacing in modern Nautilus (Ward, 1985; compare Keupp & Riedel, 1995) as well as in ammonoids (Kraft, Korn & Klug, 2008). However, there are no visible injuries on the conch of Nm. 2, suggesting that this might have not been the case. Although the ammonite could have repaired a shell injury, it would be hard to recognize the presence of such a sublethal injury due to low resolution or the effects of shell replacement. Environmental changes might also have affected the conch construction. For example, in modern scleractinian corals, it is suggested that the Mg/Ca ratio in the sea water alters the skeletal growth rate (Ries, Stanley & Hardie, 2006). The knowledge of the sedimentary facies of the host rock from which the ammonites were extracted is insufficient to identify possible causes for the alteration of shell growth. Another possibility is the presence of parasites such as tube worms. They might have grown on the external conch, which affected the buoyancy of the ammonite. Interestingly, Nm. 1 preserves the trace of a worm tube in the 41th chamber of the fifth whorl (Tajika et al., 2015), which had no detectable effect on chamber formation (Fig. 2A). Because of the absence of any trace of syn vivo epifauna on the conch, this scenario is unlikely.

The two different cumulative volumes of phragmocone chambers should result in a difference in buoyancy, given that the size of the two ammonites is more or less equal. The buoyancy of Nm. 1 was calculated by Tajika et al. (2015) as being positively buoyant in the (unlikely) absence of cameral liquid. Based on these calculations, they estimated the fill fraction of cameral liquid to attain neutral buoyancy as being about 27%. Unfortunately, the incompleteness of the aperture of Nm. 2 does not permit to calculate the buoyancy. It is quite reasonable, however, to speculate that Nm. 2 requires slightly less cameral liquid to reach neutral buoyancy (>27%) because of its size, its smaller phragmocone, and its most-likely similar conch mass. The fact that specimens with only minute morphological differences of the same species (Normannites mitis) likely expressed variation in buoyancy raises the question whether morphologically more diverse genera like Amaltheus (Hammer & Bucher, 2006) also varied more strongly in buoyancy regulation.

Ontogenetic volumetric growth of modern Nautilus and its intraspecific variation

Landman, Rye & Shelton (1983) reported that the first seven septa of Recent Nautilus are more widely spaced than the following ones; the point where septal spacing changes lies between the 7th and 8th chamber. It is considered to correspond to the time of hatching, which is also reflected in the formation of a shell-thickening and growth halt known as the nepionic constriction. This feature is also reported from fossil nautilids (Landman, Rye & Shelton, 1983; Wani & Ayyasami, 2009; Wani & Mapes, 2010).

In the growth trajectories of specimen 17 (Fig. 3A and Table 3), there are two abnormalities (the 5th and 6th chambers). These are supposedly artefacts caused by the low resolution of the scan combined with the small size of these structures and the resulting course surface of extracted chamber volume. This can occasionally cause some errors in calculating volumes in Matlab. But this problem occurred only in specimen 17, even though the low resolution would have caused errors in early rather than in late ontogeny.

Our results revealed a constant growth rate until the 5th or 6th chamber (Fig. 4B). Thereafter, the growth changes to another constant growth rate. Differences in the position of the nepionic constriction may be the artefact of low resolution of the scan, which might have made the very first (and possibly the second) chamber invisible. The position of the nepionic constriction, however, has some intraspecific variation (Chirat, 2001). Stenzel (1964) and Landman et al. (1994) showed some septal crowding between septa number 4 and 5 and between 9 and 10, respectively. Taking this into account, it is likely that our results are reflections of intraspecific variation. Nevertheless, in each examined specimen, the chamber volumes fluctuate but typically increase until the appearance of the nepionic constriction (Table 3). At the mature growth stage, most specimens show a volume reduction of the last chamber. Variability in chamber volume could be a consequence of several factors that influence the rate of chamber formation (growth rate): temperature, pH (carbon saturation degree), trace elements, food availability, sexual dimorphism, injuries, and genetic predisposition for certain metabolic features.

A relevant model for shell growth may be the ‘temperature size rule’ (e.g., Atkinson, 1994; Irie, 2010) which states the negative relationship between temperature and maturation size at moderate temperature, even though the growth rate slows down and the body size increases under extremely high or low temperatures. Rosa et al. (2012) observed a significant negative correlation between sea surface temperature and body size in coastal cephalopods. If this rule is applicable to the examined Nautilus, the temperature might have changed the growth rate of each individual because vertical migration of Nautilus is reported to range from near the sea surface to slightly below 700 m (Dunstan, Ward & Marshall, 2011). Dunstan, Ward & Marshall (2011) also suggested that the strategy for vertical migration of geographically separated Nautilus populations may vary depending on the slope, terrain and biological community. At this point, it is hard to conclude whether or not the temperature size rule applies because the behaviour of Nautilus in the Philippines can be highly different from Australian Nautilus as reported by Dunstan, Ward & Marshall (2011). Knowledge of their behaviour or possible environmental preference during growth is necessary to examine this aspect. Westermann et al. (2004) described the period of chamber formation of Nautilus pompilius which ranges from 14 to more than 400 days. It is still likely that one individual inhabited different water depths from other individuals, producing varying trends of growth trajectories. Tracking the behaviour of modern Nautilus in the Philippines may provide more information on the role and applicability of the temperature size rule.

Analyses of stable isotopes have been used to estimate habitats of shelled animals (e.g., Landman et al., 1994; Moriya et al., 2003; Auclair et al., 2004; Lécuyer & Bucher, 2006; Lukeneder et al., 2010; Ohno, Miyaji & Wani, 2015). It might be worthwhile to examine the isotopic composition of the shells of a few nautilid and ammonoid shells with different volumetric changes through ontogeny, because this may shed some light on the relationships between habitat and growth trajectories.

The pH (or carbon saturation degree) influences shell secretion. A decrease of carbon saturation causes a lack of CO32−-ions, which are required to produce aragonitic or calcitic shells (e.g., Ries, Cohen & McCorkle, 2009). In Sepia officinalis, elevated calcification rates under hypercapnic conditions have been shown by Gutowska et al. (2010). This change in pH may alter the time needed to form a chamber and thereby reduce or increase the chamber volume. Similarly, trace elements like the Mg/Ca ratio in the sea water can affect the growth rate (for corals see, e.g., Ries, Stanley & Hardie, 2006). Food availability (e.g., lack of food) is also a possible explanation for the great variation. Wiedmann & Boletzky (1982) showed a link between lack of food and lower growth rates, resulting in closer septal spacing. Strömgren & Cary (1984) demonstrated a positive correlation between growth rate of mussels and food source. It is likely that there was at least some competition for food between Nautilus individuals and probably also with other animals. The individuals in a weaker position might have had access to less food or food of poorer quality.

Intraspecific variability can also originate from sexual dimorphism. In the case of Nautilus, males tend to be slightly larger than females with slightly broader adult body chambers (Hayasaka et al., 2010; Saunders & Ward, 2010; Tanabe & Tsukahara, 2010). However, in the juvenile stage, the morphological differences are not very pronounced, thus often hampering sexing. The two average slopes in the curves of chamber volumes obtained from males and females were compared using a test (analysis of the residual sum of squares: ARSS) described in Zar (1996). This test was conducted independently for the embryonic stage and the other growth stages since the critical point between the 5th and the 6th chamber changes the slope of the growth curve (Fig. 4B). Moreover, an analysis of the residual sum of squares for nonlinear regressions was performed to compare the two average logistic models of males and females for the latter stage (Fig. 4C). No significant difference in the embryonic stage and a significant difference in the later stage (Tables 5 and 6) suggest that the differentiation in chamber volume between both sexes begins immediately after hatching. The results (Fig. 4) also show, however, the occurrence of conch morphologies common to both sexes. Taking this into account, their volume is not an ideal tool for sexing. The same statistical test for linear regressions was also conducted to compare the number of formed chambers (Fig. 5A) and the phragmocone volume (Fig. 5B) with maximum conch diameter between male and female individuals. The test results (Table 5) suggest that there is a significant difference between the female and male in both cases, although the significance levels are not strict (the number of chambers vs. maximum diameter: P < 0.05: the entire phragmocone volume vs. maximum diameter: P < 0.1). A greater sample, however, may yield a clearer separation. The results of a series of statistical tests (Table 5; analyses of the residual sum of squares) suggest that the males tend to produce more chambers than females with nearly the same conch diameter. Bearing in mind that mature males are generally larger than mature females in maximum conch diameter (Hayasaka et al., 2010), this may potentially indicate a prolonged life span or less energetic investment in reproduction. By contrast, the addition of another chamber to males could be associated with their sexual maturity; the weight of the large spadix and a large mass of spermatophores in males might necessitate more space and buoyancy. Ward et al. (1977) reported that the total weight of males of Nautilus pompilius from Fiji exceeds that of females by as much as 20%. What remains unclear is the reason why females tend to have larger phragmocone volumes than males while they are immature. It is true, however, that even within each sex, the variability of the total phragmocone volumes is quite high (standard deviation for males: 15.4; for females: 13.4; for both males and females: 14.3). Of course one should also bear in mind the possibility that these high variabilities may have partially originated from the errors discussed in the method section.

Injuries are visible in several of the examined specimens, yet there is no link to a temporal or spatial change in chamber volume in the growth curves. Yomogida & Wani (2013) examined injuries of Nautilus pompilius from the same locality in the Philippines, reporting traces of frequent sublethal attacks rather early in ontogeny than in later stages. The frequency of sublethal attacks early in ontogeny may be one of the factors determining the steepness of the growth trajectory curves. This aspect can be tested in further studies. Additionally, morphological variability may also root in genetic variability but the causal link is difficult to test.

Covariation of chamber volumes and widths in ammonoids and nautiloids

The relationship between chamber volumes of Nautilus pompilius (Fig. 7) revealed that their chamber widths expanded at a constant pace irrespective of the change in chamber volume. For the construction of the Nautilus conch and its ontogeny, a rather constant conch morphology might have been advantageous with the buoyancy regulation depending largely on septal spacing only. Likewise, Hoffmann, Reinhoff & Lemanis (2015) reported on Spirula that has a sudden decrease of chamber volume which is not correlated with changes in whorl height or whorl width but with changes in septal spacing. By contrast, the chamber widths and volumes of the ammonites appear to covary (Fig. 6). This distinct covariation may have partially contributed to the high morphological variability with some constraints in response to fluctuating environmental conditions or predatory attacks (for details, see the discussion for Nautilus above). This aspect, however, needs to be examined further using an image stack of an ammonite with a higher resolution and better preservation to rule out artefacts.

Conclusions

We virtually reconstructed the conchs of two Middle Jurassic ammonites (Normannites mitis) and 30 specimens of Recent nautilids (Nautilus pompilius) using grinding tomography and computed tomography (CT), respectively, to analyse the intraspecific variability in volumetric change of their chambers throughout ontogeny. The data obtained from the constructed 3D models led to the following conclusions:

1. Chamber volumes of Normannites mitis and Nautilus pompilius were measured to compare the ontogenetic change. The growth trajectories of Normannites mitis and Nautilus pompilius follow logistic curves throughout most of their ontogeny. The last several chambers of Normanites mitis show fluctuating chamber volumes, while most specimens of Nautilus pompilius demonstrate a volume reduction of only the last chamber.

2. Growth trajectories of the two Normannites mitis specimens were compared. The two specimens appear to have a transition point between the 28th and 29th chamber from which the slopes of their growth curves change, which has been documented previously. However, their entire phragmocone volumes differ markedly in late ontogeny although the two shells have similar morphology and size. Intraspecific variation of buoyancy was not testable in this study due to the low sample number. This aspect needs to be addressed in future research because buoyancy analyses could provide information on the habitat of ammonoids.

3. Growth trajectories of thirty Nautilus pompilius conchs show a high variability, even though the high variabilitiy may have partially originated from the errors discussed in the method section.

4. Results of statistical tests for Nautilus pompilius corroborate that the variability is increased by the morphological difference between the two sexes: adult males have larger volumes than females with the same diameter. This may be ascribed to the formation of voluminous reproductive organs in the male (spadix). Individual chamber volumes of the female tend to be larger than those of males. The results also show that intraspecific variability within one sex is moderately strong. Examinations of their injuries, isotopic analyses of the examined conchs or tracking the behaviour of Nautilus could yield more information on the relationship between their variability in chamber volumes and ecology. Such data could help to reconstruct the palaeoecology of fossil nautiloids and possibly also of extinct ammonoids.

5. Covariation between the chamber widths and volumes in ammonites and Nautilus pompilius were examined. The results illustrate that conch construction of Nautilus pompilius is robust, maintaining a certain shape, whereas the conch development of the examined ammonites was more plastic, changing shape during growth under some fabricational constraints. Further investigations need to be carried out to verify the covariation between widths and volumes of ammonites with other variables such as conch thickness, conch width, and perhaps buoyancy using a reconstruction method with a higher resolution and perfectly-preserved materials.

Supplemental Information

Table S1 Estimated errors resulting from CT scans in Nautilus pompilius

Actual shell volumes were calculated based on measurement of weight of specimens with possible minimum shell density (2.54 g/cm3; Hoffmann & Zachow, 2011) and maximum shell density (2.62 g/cm3; Reyment, 1958) of Nautilus.

Click here for additional data file.

We would like to thank Dominik Hennhöfer and Enric Pascual Cebrian (Universität Heidelberg) for carrying out the grinding tomography. Beat Imhof (Trimbach) kindly donated the two specimens of Normannites. We are also thankful to Torsten Scheyer (Universität Zürich) for the introduction to the use of Avizo® 8.1. Kathleen Ritterbush (University of Chicago) proofread the manuscript and corrected the English. The fruitful discussion with Kozue Nishida (The Geological Survey of Japan) is greatly appreciated.

Additional Information and Declarations

Competing Interests

Author Contributions

The authors declare there are no competing interests.

Amane Tajika conceived and designed the experiments, performed the experiments, analyzed the data, wrote the paper, prepared figures and/or tables, reviewed drafts of the paper.

Naoki Morimoto performed the experiments, contributed reagents/materials/analysis tools, reviewed drafts of the paper.

Ryoji Wani and Carole Naglik contributed reagents/materials/analysis tools, reviewed drafts of the paper.

Christian Klug wrote the paper, reviewed drafts of the paper.

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
