# Peer review of "Intraspecific variation of phragmocone chamber volumes throughout ontogeny in the modern nautilid Nautilus and the Jurassic ammonite Normannites"

_PeerJ, doi:10.7717/peerj.1306_

## Round 0.1 · original submission · Major Revisions

Dear authors,

I have now received two reviews for your manuscript entitled, Intraspecific variation of phragmocone chamber volumes throughout ontogeny in modern Nautilus and the Jurassic ammonite Normannites. Both reviewers think the paper is an important, data-rich study that will make a valuable contribution to the literature on variation in conch volume and geometry, with considerable broader implications for palaeobiology and palaeoecology. I agree with their positive comments, and I think this paper will make a valuable addition following some revisions.

Overall, Reviewer #2 (open review: Mathew Knauss) has made some minor, helpful points, particularly the suggestion for a summary figure or additional paragraph in the discussion to better contextualize the results in terms of previous studies on intraspecific variation would be a valuable enhancement to the paper. Reviewer #1 (open review: Robert Lemanis) has listed some slightly more extensive comments that the authors should pay careful attention to in a revised version. Particularly, please note suggestions made to improve clarity of the methods (regarding scanning parameters, and additional references). Reviewer #1 also suggests the authors may wish to calculate the correlation coefficient between width and volume (see comment line 339), and has flagged up several queries regarding the conclusions drawn from the data presented. Please address those fully. Finally, Reviewer #1 has made some suggestions regarding improvements to the graphs presented in several figures, these will be helpful to the reader and should be quick to resolve (consistent use of symbol color/shape for each sex across graphs, for example).

I have added some minor editorial suggestions in the attached pdf, please also take care to check that all citations in the text are listed in the reference section of the paper.

I would look forward to receiving a revised version of your paper in the near future.

·

Basic reporting

Overall the structure of the manuscript is clear. Some improvements can be made such as moving the description of the statistical analysis from the discussion to the results section. Improvements regarding the clarity and structure of the figures can be made, these are elaborated in the comments provided on the annotated manuscript.
Some references cited in the text are not in the reference list and some additional useful references are suggested in the attached manuscript.

Experimental design

Hypothesis to be tested are clearly stated in the introduction. However, further information should be added to the methods section that is important in both the evaluation of the data presented and reproducibility of the data.
There are some significant misconceptions about some of the methods used in the paper that need to be addressed. Furthermore significant sources of error are not adequately addressed and presented.

Validity of the findings

As mentioned above, misconceptions about sources of error are present and therefore these errors are not included in a discussion of the data. It is difficult to say how this affects the conclusions.
Some of the presented conclusions don’t seem to follow from the actual presented data. Some concluding statements seem somewhat contradictory to other statements in the manuscript (these are commented upon in the annotated attachment).

Additional comments

Overall I think this is a good paper and an important study. Variability in volumetrics is an important topic to address and the sheer number of specimens you worked on is very impressive. That being said there are several significant misconceptions presented in the paper and I would like to see some additional data presented. With significant revisions this can be a very interesting paper especially given the fact that the application of quantitative tomographic techniques to cephalopods is a new field.

·

Basic reporting

Overall, I would conclude that this is a great contribution to science. The study provides an excellent, concise background to the issue at hand: accurately calculating buoyancy in the conchs of cephalopods, which has implications for understanding ontogeny, inter/intraspecific variation, and paleoecology of extinct groups. The authors chose a large sample size of Nautilus in which to study, and while the sample size for Normannites is less than preferable for any field of science, the authors explain the issues encountered when trying to generate three-dimensional models from tomographic datasets (e.g., sufficient density contrast, time necessary for processing, etc.).

I have some corrections to the text that would make this submission more polished and will be easy for the authors to incorporate.

Line 92: The "," after the "and" is not necessary.
Line 93: You are missing a word: "...application OF this method for recent Nautilus."
Line 100-101: Incorrect subject-verb agreement: "Volumetric analyses...HAVE not previously been analyzed..." (change "has" to "have").
Line 135: You might need a citation for Adobe Illustrator.
Line 136: You might need a citation for VGstudiomax 2.1.
Line 144: Add a space between the value and its units, such that it says "0.2 mm."
Line 146: You might need a citation for Avizo 8.1 (I believe it is FEI Visualization Sciences Group).
Line 158: You should cite Meshlab (I believe it is ISTI- CNR research center).
Line 206: You are missing the last word of the sentence; "...were negligible for our STUDY (Fig. 6B)."
Line 245: I would change "...and the probably nearly identical conch mass" to something like this: "....and ITS most-likely identical conch mass." Changing "the" to "its" creates consistency for each thing listed and replacing the dual adverb "probably nearly" with something like "most-likely" is less awkward to read.
Line 267: You might have an extra space between "which" and "states." I would just check that.
Line 331: Change "grow" to "growth."
Line 359: I think you are missing a word. Perhaps, just change "in previous" to just "previously" to make it more concise.
Line 426: In the Gotz, 2003 citation, you need a space in "GeolCroat" so that it becomes "Geol Croat."
Line 503: In the Tanabe and Tsukahara, 2010 citation, you have a typo. Change "Biometrie" to "Biometric."

Experimental design

No Comments

Validity of the findings

Overall, the conclusions made by the authors are consistent with the results and figures. The discussion of the manuscript hits all the major points that could explain the results. However, it would be nice to see a more in depth discussion on how the results presented in this manuscript relate to those found in previous studies regarding interspecific variation. One way to do that may involve creating an additional figure that summarizes the results of this study along with those of Naglik et al., 2015a and 2015b (or any other related study, if permitted). While these studies really don't hit intraspecific variation, there is still much that could be said about the overall trends between ammonoids and Nautilus.

Additional comments

No comments.

---

## Round 0.2 · Minor Revisions

Dear Authors,

Thank you for carefully revising your earlier manuscript in line with comments from the reviewers. Reviewer #1 considers the revised version to be much improved and almost ready for publication pending some minor revisions. I agree with the Reviewer's comments, and ask if you could please follow their suggestion to include more details of the percentage error in volume estimation in the text. In re-reading your paper I have very few additional comments, beyond a couple of minor wording suggestions, below:

Pg4, ln57: “Now complete..” this sentence is a fragment and needs to be reworded
Pg6, Ln86: suggest adding “process” after “segmentation”
Pg9, ln165: I would rephrase, “But of course”
Pg17, ln333: “food availability is also a possible explanation”. This is vague and could be rephrased to add clarity to this sentence – i.e. by stating “lack of food”. Otherwise, this sentence could be deleted.
Pg 18, ln356: in favour of economy, “appear to imply” could be replaced by “suggest”

·

Basic reporting

I think this version of the manuscript is clearly improved from the previous version. I have included an annotated pdf with some minor corrections to language and comments.

Experimental design

Experimental questions have been clearly stated and addressed in the text.

Validity of the findings

The supplemental table detailing the error estimates for Nautilus are much appreciated. However the range of this error in volume should be more clearly stated in the text. The error is up to as much as about 90% which means that a good portion of the variability in chamber volume for any given chamber may be largely explained by error. The readers should be made aware of this more clearly.

If possible, it would also be nice if the two Normanittes specimens can be subjected to the same treatment. Of course you can't measure the actual weight or volume of the shell but compare the reconstructed volume with the volume of the same labelfield/stl that has been expanded by a voxel layer. This is not as reliable as what you did with Nautilus but it can give us an idea about the sensitivity of the data to segmentation and resolution.

---

## Round 0.3 · accepted · Accept

The authors are thanked for addressing all the minor wording points raised in the last round of reviews in a timely manner, and for a thorough justification of their approach to calculating volume measures. I think these changes have now fully addressed all comments to the manuscript and that this version is ready for publication. I look forward to seeing your article published.